# Utilizing Gold Nanoparticle Decoration for Enhanced UV Photodetection in CdS Thin Films Fabricated by Pulsed Laser Deposition: Exploiting Plasmon-Induced Effects

**DOI:** 10.3390/nano14050416

**Published:** 2024-02-24

**Authors:** Walid Belaid, Serap Yiğit Gezgin, Mohamed A. Basyooni-M. Kabatas, Yasin Ramazan Eker, Hamdi Şükür Kiliç

**Affiliations:** 1Department of Physics, Faculty of Science, Selçuk University, Konya 42075, Turkey; walid.belaid@usmba.ac.ma (W.B.);; 2Dynamics of Micro and Nano Systems Group, Department of Precision and Microsystems Engineering, Delft University of Technology, Mekelweg 2, 2628 CD Delft, The Netherlands; 3Solar Research Laboratory, Solar and Space Research Department, National Research Institute of Astronomy and Geophysics, Cairo 11421, Egypt; 4Science and Technology Research and Application Center (BITAM), Necmettin Erbakan University, Konya 42090, Turkey; 5Department of Basic Sciences, Faculty of Engineering, Necmettin Erbakan University, Konya 42090, Turkey; 6Directorate of High Technology Research and Application Center, University of Selçuk, Konya 42031, Turkey; 7Directorate of Laser-Induced Proton Therapy Application and Research Center, University of Selçuk, Konya 42031, Turkey

**Keywords:** UV sensors, Au nanoparticles, CdS thin films, photodetection, pulsed laser deposition

## Abstract

UV sensors hold significant promise for various applications in both military and civilian domains. However, achieving exceptional detectivity, responsivity, and rapid rise/decay times remains a notable challenge. In this study, we address this challenge by investigating the photodetection properties of CdS thin films and the influence of surface-deposited gold nanoparticles (AuNPs) on their performance. CdS thin films were produced using the pulsed laser deposition (PLD) technique on glass substrates, with CdS layers at a 100, 150, and 200 nm thickness. Extensive characterization was performed to evaluate the thin films’ structural, morphological, and optical properties. Photodetector devices based on CdS and AuNPs/CdS films were fabricated, and their performance parameters were evaluated under 365 nm light illumination. Our findings demonstrated that reducing CdS layer thickness enhanced performance concerning detectivity, responsivity, external quantum efficiency (EQE), and photocurrent gain. Furthermore, AuNP deposition on the surface of CdS films exhibited a substantial influence, especially on devices with thinner CdS layers. Among the configurations, AuNPs/CdS(100 nm) demonstrated the highest values in all evaluated parameters, including detectivity ($1.1\times 10^{12}$ Jones), responsivity (13.86 A/W), EQE (47.2%), and photocurrent gain (9.2).

## 1. Introduction

The demand for highly efficient and responsive UV sensors has grown significantly in recent years due to their wide-ranging applications in the military and civilian sectors. UV sensors are vital in various fields, including environmental monitoring, industrial processes, and biomedical diagnostics. Ultraviolet (UV) radiation, while instrumental for its antibacterial properties that promote human health by eradicating pathogenic microorganisms, can pose significant health risks. Prolonged exposure to UV radiation is associated with detrimental health outcomes such as an increased risk of skin cancer, the development of cataracts, and the suppression of immune system functions. This duality underscores the importance of accurately monitoring UV radiation levels, a task for which UV sensors are essential [1,2,3]. These sensors transform UV light into electrical signals, a process facilitated by photodiodes or photoresistors. Photodiode-based UV sensors are characterized by their ability to generate an electric current in response to light exposure. This current, proportional to the intensity of the incident light, is converted into a measurable output, either digital or analog, reflecting the level of UV light.

In contrast, photoresistor UV sensors operate based on the principle of light-induced resistance change. While these sensors offer less precision than their photodiode counterparts, they are suitable for applications requiring essential light detection, such as determining the presence or absence of light or gauging relative light intensities in specific environments [4,5]. However, achieving exceptional responsivity and rapid response/recovery times remains a noteworthy challenge. This has motivated extensive research efforts to explore novel materials and device architectures that can enhance the performance of UV sensors [6,7].

CdS has emerged as a promising semiconductor material for optoelectronic devices and detectors thanks to its exceptional properties, such as a direct bandgap (∼2.4 eV), excellent transport characteristics, and high refractive index [8,9,10,11,12,13,14]. The performance of CdS thin film-based devices is influenced by film thickness, grain size, and morphology [15,16]. CdS thin films have been extensively explored for their optoelectronic properties, and various nanostructures such as thin films [17], nanowires [18], nanobelts [19], and nanoribbons [20] have been developed to improve their photodetector performance. Various fabrication methods such as sputtering [21], vacuum evaporation [22], electrodeposition [23], chemical bath deposition [24,25,26], spray pyrolysis [27,28], chemical vapor deposition [29], metalorganic chemical vapor deposition (MOCVD) [30], and pulsed laser deposition (PLD) [31,32,33] have been employed to produce high-quality CdS thin films. In recent years, the PLD technique has garnered significant attention for delivering sophisticated, high-quality compound semiconductor thin and ultrathin films and embedding nanoparticles (NPs). In this technique, the stoichiometry of the target is preserved in the growing film due to the fast temperature rise (>10^11^ K s^−1^) produced by a high-power laser that ablates each component of the target materials simultaneously [34]. With a highly well-regulated deposition rate, the composition of the used target is promptly transmitted to the substrate, which is a significant benefit of the PLD system [35,36]. Furthermore, in PLD, thin film formation can occur under various reactive conditions, such as different ambient gases or vacuum [32], which can influence film characteristics, although the specific effects of these conditions were not the primary focus of our current investigation. Other factors to consider are laser power and frequency [37], ambient gas pressure [38], and substrate temperature [39]. Our group has made notable contributions to this field with a series of studies [40,41,42].

Moreover, the development of nanomaterials, particularly NPs, has garnered significant attention due to their distinct properties and potential for enhancing various applications. NPs offer unique characteristics that differ from their bulk counterparts, making them highly promising for optoelectronic devices. In particular, metallic NPs have been extensively studied for their localized surface plasmon resonance (LSPR) effect, which can significantly enhance the optical properties of materials. The LSPR phenomenon, arising from the collective oscillation of conduction electrons in NP, allows for efficient light absorption and scattering. This property has been successfully harnessed to enhance the performance of photodetection devices, offering new avenues for achieving improved responsivity, sensitivity, and overall device performance [37,43]. In line with these advancements, this study explores the incorporation of Au nanoparticles (AuNPs) on CdS thin films to leverage the LSPR effect and enhance the photodetection properties of the devices.

Our review of the existing literature has identified individual studies that address the influence of film thickness on photodetector properties, notably by Makhdoumi et al. [44], as well as investigations into the effects of doping on the photosensing characteristics of CdS photodetectors [9,45,46]. Despite these contributions, our analysis indicates a lack of comprehensive research that synergistically examines the effects of both plasmonic material deposition and film thickness on CdS photodetectors. In particular, the interdependent impact of plasmonic material deposition in conjunction with variations in CdS film thickness on photosensing performance has not been thoroughly explored. Therefore, this study aims to bridge this gap by providing a detailed investigation into how these combined factors influence the photosensitive properties of CdS photodetectors. In light of this, this study focuses on developing high-quality CdS/glass and AuNPs/CdS/glass nanostructure thin films with different thicknesses (100, 150, and 200 nm) of the CdS layer. These films were fabricated employing the PLD technique on glass substrates. In the ambit of this investigation, a systematic nomenclature has been adopted for the various samples under examination to facilitate clarity and precision in discourse. Specifically, cadmium sulfide (CdS) thin films with 100 nm, 150 nm, and 200 nm are designated as S1, S2, and S3, respectively. Correspondingly, the samples comprising gold nanoparticles (AuNPs) deposited on CdS thin films of identical thicknesses are labeled as S4 (100 nm), S5 (150 nm), and S6 (200 nm). The morphology, structure, and optical/photoluminescence properties of thin films were thoroughly examined. CdS-based photodetector devices were fabricated, and their performance characteristics, including I-V characteristics, response/recovery time, responsivity (R), photosensitivity, detectivity (D*), and external quantum efficiency (EQE), were systematically evaluated in correlation with specific parameters. Therefore, the investigation focused on analyzing the impact of varying CdS thin film thickness and the influence of Au nanoparticles (AuNPs) on their photodetection performance. Detailed discussions regarding these aspects were provided to gain a comprehensive understanding of the photodetection behavior. Through this study, we establish a nuanced understanding of the interplay between nanoscale architecture and photodetector efficacy, offering valuable insights that may guide future innovations in photodetector technology.

## 2. Materials and Methods

### 2.1. Materials

The primary materials used to fabricate thin films during this study included cadmium sulfide (CdS) and gold (Au), which were integral to our experimental setup. The CdS target, essential for developing the CdS thin films, was a disc of 5 cm diameter with a purity of 5N (99.999%), sourced from Sigma Aldrich (St. Louis, MO, USA). This target was employed to deposit CdS layers onto pre-cleaned glass substrates. Similarly, for the deposition involving gold, a 5 cm diameter target of gold with a purity of 5 N was procured from Sigma Aldrich. Silver (Ag) Conductive Adhesive Paste, obtained from Nanografi (Ankara, Türkiye), was also used to establish electrical contacts. The actual device in our study is based on two parallel electrodes designed from Ag that are 3 mm in width and 1 cm in length. Meanwhile, the total area of the sample has the dimensions of 1 × 1.5 cm^2^.

### 2.2. Deposition Technique

Thin films were grown using a custom-built PLD system developed by our group [47], in combination with a Continuum Minilite II Nd: YAG Laser (Singapore) delivering photons at a fundamental 1064 nm wavelength operating in pulsed mode with five ns pulse duration at a 10 Hz repetition rate as in Appendix A. A neutral density filter regulated the laser energy. Substrate and target were placed on rotor-controlled rotating holders to ensure homogeneous plasma generation and avoid substrate damage. This resulted in a uniform thin film growth. All experiments were performed at ambient temperature in a vacuum chamber with a 5 × 10^−7^ mbar background pressure. The system design is described in our previous work [37]. In the deposition technique employed for both CdS and AuNPs/CdS thin films, the thickness of the films was precisely controlled by modulating the duration of the deposition process, ensuring specific and uniform film characteristics as per our experimental requirements. The sample thickness was verified using a Filmetrics F20 Thin Film Analyzer (Filmetrics, San Diego, CA, USA), demonstrating a deviation of approximately 1 nm.

### 2.3. Synthesis of CdS Thin Films

The synthesis of CdS thin films involved the ablation process of the target material by laser irradiation, utilizing a cadmium sulfide (CdS) target. The laser beam’s energy per pulse was carefully controlled at 12 mJ at the filter–lens interface to ensure optimal ablation. Before the deposition process, the distance between the target and the substrate was meticulously set to 75 mm.

### 2.4. Synthesis of AuNPs/CdS Thin Film

The synthesis of AuNPs involved the ablation process of the gold target by laser irradiation on the previously produced CdS thin films. The deposition process of AuNPs thin films on glass substrates using PLD was once diagnosed/optimized and reported [37,38]. In this investigation, AuNPs were deposited onto CdS samples using a 20 mJ laser beam with a 20 min exposure duration. Before experimentation, the target–substrate distance was set to 50 mm during AuNPs deposition.

### 2.5. Device Fabrication

The silver paste was used to synthesize metallic contacts on the front side of the fabricated thin films (See Appendix A). The samples of Ag/(S1 or S2 or S3)/glass and Ag/(S4 or S5 or S6)/glass devices are covered in this study. AuNPs/S1, AuNPs/S2, and AuNPs/S3 are denoted as S4, S5, and S6, respectively.

### 2.6. Characterization

The structural and phase parameters of CdS and AuNPs/CdS films were analyzed using an X-ray diffractometer (Panalytical Empyrean, Malvern Panalytical, Worcestershire, UK) equipped with CuKα radiation (λ = 1.5405 Å, 40 kV, 100 mA). The device was set with a step size of 0.01 and a 2θ range of 10°–80° to perform the analysis. A ZEISS GeminiSEM 500 (Oberkochen, Germany) field emission scanning electron microscope (FE-SEM) with an Electron High Tension (EHT) of 1.00 Kv and a Working Distance (WD) of 4.6 nm was employed to analyze surface morphology. The optical absorption spectra were measured using a UV-Vis spectrophotometer (JASCO, V-670 Spectrophotometer, Tokyo, Japan) covering a wavelength range from 200 to 2000 nm. Electrical current–voltage (I–V) and optoelectronic measurements were conducted using a source meter under a 365 nm UV light lamp, enabling the assessment of the electrical properties of the thin film sample. The light was directed towards the device’s front side, as illustrated in the schematic diagram presented in Figure 1. Photoresponse measurements were taken under a bias voltage of 5.0 V with an ON-OFF cadence of 1 s for 12 s.

## 3. Results and Discussions

### 3.1. XRD Analysis

According to the XRD pattern illustrated in Figure 2, it is evident that all CdS and AuNPs/CdS thin films exhibit a polycrystalline structure at room temperature without the application of heat treatment. The diffraction peaks identified at 2θ = 24.91°, 26.65°, 28.29°, 36.70°, 43.77°, and 62.73° angles correspond to the (100), (002), (101), (102), (110), and (104) planes, respectively, indicating a hexagonal phase structure of the thin films. Among these, the (102) and (110) orientations exhibit higher peak intensities, suggesting a preferred directional growth of particles along these planes due to their lower surface energy, a phenomenon that is in agreement with the existing literature [48,49,50,51,52]. This preferred orientation is attributed to the stacking sequence of close-packed Cd planes, with S atoms occupying tetrahedral interstitial sites, emphasizing the significance of these dominant orientations in the crystal growth process [53]. Enhancing our understanding further, the bombardment of the CdS surface with gold nanoparticles, mainly observed in the thinner CdS thin film (sample S4), demonstrates a unique interaction. This process facilitates a rearrangement of surface atoms, seeking stable states through forming additional bonds, which is more pronounced in samples with enhanced surface-to-volume ratios like S4. The zoomed-in view of the XRD pattern focusing on the (102) direction (Figure 2b) showcases differences in peak intensity and crystallinity among the samples, with S4 exhibiting the best crystalline quality.

The crystalline sizes of thin films are calculated with the following Scherrer equation:(1)D=0.94λ/βcosθ
where the D, λ, β, and θ parameters are the crystalline size, the wavelength of X-ray, the fullwidth at half-maximum of diffraction peak, and the Bragg diffraction angle, respectively. Table 1 presents the XRD characteristics of CdS and AuNPs/CdS thin films (samples S1–S6), including the values for Full Width at Half Maximum (FWHM) and the average values for crystallite size (D) and crystallite density (δ). Our analysis aligns with the methodology outlined in Rahmi et al.’s study [54]. According to XRD, Au doping improved the crystal structure of the CdS thin film. In particular, the crystal structure of the S1 was enhanced by Au doping. So, Au atoms replaced Cd atoms since the ionic radius of Au^+^(151 Å) is more significant than that of Cd^2+^ (0.97 Å), and the size of the crystal has increased [37,55,56,57]. As the grain size increased, the number of grain boundaries decreased. Thus, it is passivized by the trap located between the grain boundaries. This enhanced the diffusion of minority charge carriers and increased the lifetime. Therefore, photoexcited charges contribute to the performance of the photodetector while not being affected much by defects and traps [58,59,60,61].

The crystalline density of thin films [62,63] is determined by Equation (2):(2)$$\delta =\frac{1}{D^{2}}$$

According to Table 1, sample S4 contains the lowest dislocation density because the defects and traps in the ultrathin film were slightly reduced by depositing Au nanoparticles onto the surface of the S1 thin films [64].

### 3.2. FESEM Surface Morphology Analysis

This study has used the PLD technique to prepare thin films based on CdS (cadmium sulfide) and AuNPs/CdS (gold nanoparticles/cadmium sulfide). These films were prepared at different thicknesses and characterized their surfaces using field-emission scanning electron microscopy (FESEM), as shown in Figure 3. FESEM images revealed detailed and high-resolution morphology of thin films, showing densely distributed particles. Our quantitative analysis using ImageJ 1.52a has revealed a significant presence of highly organized nanoparticles on the CdS film surface in Appendix A. On the surface of the CdS film, it has been shown that highly organized NPs cover the entire surface. All thin films exhibited small grains with spherical particles (indicated by arrows). However, due to differences in surface energy, some particles tended to cluster together or agglomerate. The surface particles were more prominent in the case of S1, which had a low thickness and a relatively short deposition period. As deposition periods increased to S2, an additional material layer filled the porous surface areas. A similar behavior was observed for S6, where the thickness of the film was even greater. Interestingly, S1 showed smaller particles on the surface, which could potentially enhance its light-harvesting capabilities compared to the thicker thin films.

When AuNPs (gold nanoparticles) were deposited, it was noticed that the surface morphology of sample S4 appeared smoother. We did not observe large agglomerations; the particles were well distributed across the entire surface. Similar effects were also observed for S5 and S6 samples. It is worth mentioning that the thickness of AuNPs remained constant throughout the deposition process, which was achieved by using a rotating substrate holder to ensure better uniformity. The formation of small agglomerations can be attributed to the fact that the substrate temperature was maintained at room temperature during the deposition. These observations align with similar effects that have been previously reported [37]. While the materials differ, the morphological deposition-induced impact, such as grain coalescence and surface texturing due to secondary phase formation, exhibit analogous behavior.

### 3.3. Optical Analysis

Understanding the electronic structure of CdS thin films is crucial for investigating the photodetection mechanism exhibited by these films. The absorbance spectra shown in Figure 4 include samples S1–S6. The presence of electron–phonon or exciton–phonon interactions may contribute to the appearance of an absorption edge at the bandgap energy. Notably, all film compositions demonstrate interband transitions at around 530 nm, with a blueshift observed in the band edge as CdS layer thickness increases. Additionally, it is noteworthy that thinner samples, such as S1 and S4, exhibit higher absorbance in UVA regions. This high absorbance in the UVA range is particularly significant for UV photodetection applications, as it enhances the sensitivity of thin films to detect and respond to ultraviolet radiation. Moreover, incorporating a refractive metallic Au layer causes only a minimal decrease in absorption, indicating the plasmonic effect of Au, which can be further exploited to enhance the overall photodetection performance.

The bandgap energy gap ($E_{g}$) of the films was determined using the Tauc equation $\left(\alpha h\upsilon \right)=A{\left(h\upsilon -E_{g}\right)}^{n}$. The resulting plots illustrating the energy gaps are presented in Figure 5. By extrapolating the linear portion of the plots on the *x*-axis, the energy gaps of thin films were estimated to be approximately 2.24 eV, 2.45 eV, and 2.46 eV for S1, S2, and S3, respectively. Similarly, the energy gaps for S4, S5, and S6 were approximately 2.19 eV, 2.45 eV, and 2.46 eV. The band gap of thin films increases with an increase in the CdS layer’s thickness.

Furthermore, in the case of S4, the bandgap was narrower compared to S4. This phenomenon can be ascribed to the influence of Au nanoparticles, where their presence induces higher impurity levels within the valence band, consequently causing a reduction in the bandgap. This reduction is primarily associated with the surface plasmon resonance (SPR) effect [65,66]. However, for thicker CdS films, the impact of SPR on the bandgap is negligible.

### 3.4. Photoelectrical Properties

The photodetection performance of CdS and AuNPs/CdS configurations (samples S1–S6) was thoroughly investigated by characterizing their current–voltage (I–V) features in dark and UV illumination. The samples were exposed to light from a UV halogen lamp with 365 nm, and the I-V features were recorded from −5 to +5 volts. Figure 6a,b showcase the I-V behavior of the S1–S6 photodetectors in dark and illuminated conditions. Notably, all films exhibited remarkable UV light detection capabilities.

The measured dark currents ($I_{d}$) at a bias voltage of 5.0 V were found to be approximately 3.97 µA, 1.94 µA, 0.98 µA, 2.67 mA, 2.06 µA, and 1.25 µA for the S1, S2, S3, S4, S5, and S6 thin films, respectively. Notably, the dark current exhibited a significant improvement when the thickness of the CdS layer decreased. Furthermore, the influence of AuNPs on the thinner CdS layer (S1) resulted in a notable enhancement in the dark current. Specifically, the S4 sample demonstrated the highest level of I_d_, reaching approximately 2.67 mA at a bias voltage of 5.0 V.

The time-dependent photoresponse of Ag/CdS/Ag and Ag/AuNPs/CdS/Ag photodetector devices (samples S1–S6) to UV light pulses (0.3 mW/cm^2^) was investigated to examine the temporal behavior. The photoresponse of the devices was analyzed for one cycle at 5.0 V, as depicted in Figure 7a,b. The rise and decay times were determined to be 61–85 ms and 65–87 ms, respectively. Notably, the S1-based device exhibited a faster temporal photoresponse than the S2- and S3-based photodetector devices. This can be attributed to the increased thickness of CdS thin films, which results in a longer transit time for the charge carriers to reach the electrodes.

Conversely, incorporating plasmonic NPs in the thinner CdS layer (S1) led to a notable enhancement in the response speed. Specifically, sample S4 demonstrated shorter rise and decay times. This enhancement can be attributed to hot carriers on the Fermi surface of Au NPs. However, thicker CdS layers, including Au NPs, reduced the overall photoresponse of the device. This reduction can be attributed to the localized surface plasmon resonances generated by the plasmonic nanoparticles, which enhance light absorption and increase charge carrier recombination near the nanoparticle surfaces. The increased recombination rate ultimately diminishes the overall photoresponse of the device.

The photodetector parameters such as responsivity (R), detectivity ($D^{*}$), external quantum efficiency (EQE), and photocurrent gain (G) play a crucial role in understanding the influence of CdS layer thickness and the deposition of AuNPs on the CdS layer. These parameters were calculated using the following expressions [67,68,69]:(3)$$R=\frac{I_{Ph}}{P\times S}$$
(4)$$D^{*}=R\sqrt{\frac{S}{2eI_{d}}}$$
(5)$$EQE=R\frac{hc}{e\lambda }$$
(6)$$G=\frac{I_{Ph}(I_{ON})}{I_{d}\;(I_{OFF})}$$

In Equations (3)–(6), the variable P signifies the power output of the illumination source. The symbol S is employed to denote the active surface area of the device under investigation. λ denotes the illumination wavelength, which is precisely 532 nm. The constants c, h, and e correspond to the fundamental physical quantities: the speed of light in a vacuum, Planck’s constant, and the elementary charge, respectively. Within this framework, $I_{d}$ and $I_{Ph}$ are designated to represent the dark current and the photocurrent, respectively.

The results of these parameters were obtained under a bias voltage of 5.0 V and are presented in Figure 8a. Remarkably, CdS layer thickness and the deposition of AuNPs on the CdS layer exhibited similar effects on all evaluated parameters.

The investigated parameters exhibited an increasing trend as the CdS layer thickness decreased. The impact of AuNPs on these parameters was found to be dependent on the thickness of the CdS layer. In the case of the thinner film (S1), the presence of AuNPs resulted in a notable enhancement in all evaluated parameters. However, as the thickness of the CdS layer increased, the effect of AuNPs became less remarkable. Consequently, the S4 configuration demonstrated the highest values across all studied parameters, including a responsivity of 13.86 A/W, a detectivity of $1.1\times 10^{12}$ Jones, an EQE of 47.2%, and a photocurrent gain of 9.2.

The rise and decay times of the photodetectors (PDs) S1 to S6, operating at a bias of 5.0 V, are illustrated in Figure 8b. The rise $\tau_{R}\;$(response) time holds significant importance in photodetectors as it signifies the duration required for the device to detect variations in light intensity and generate an output signal. $\tau_{R}\;$is particularly critical in applications that demand swift detection and response, such as optical communication systems and high-speed imaging. A shorter $\tau_{R}\;$enables the photodetector to rapidly respond to changes in light intensity, facilitating faster and more accurate measurements. Conversely, the decay $\tau_{D}\;$(recovery) time denotes the period required for the photodetector to return to its original state after exposure to a light signal. The reduction in CdS layer thickness leads to a decrease in both the response and recovery times. The incorporation of AuNPs has a substantial effect on reducing $\tau_{R}$ and $\tau_{D}$, especially in the case of the thinner CdS film (S1). However, as the CdS layer thickness increases, the influence of AuNPs becomes less significant. Consequently, sample S4 exhibits faster rise ($\tau_{R}$ = 61.68 ms) and decay ($\tau_{D}$ = 75.74 ms) times, aligning with the expected outcomes.

Previous studies have highlighted the importance of developing cost-effective and uniformly deposited CdS thin films to enhance the performance of photodetector devices. A comprehensive examination of the existing literature on photodetector devices based on CdS materials is presented in Table 2. However, this study proposes a novel approach to address these challenges by utilizing a simple pulsed laser deposition (PLD) system. This method enables the fabrication of CdS thin films with a surface-deposited plasmonic layer of Au nanoparticles (AuNPs). Notably, the focus is specifically on the thinner CdS layers, as they exhibit remarkable improvements in the performance of photodetectors (PDs). Consequently, this approach holds significant potential for enhancing the overall performance of PD devices.

Our investigation highlights film thickness’s critical role in defining CdS thin films’ photodetection properties. The balance between light absorption, carrier dynamics, and surface effects is pivotal in optimizing photodetector performance. This study contributes to a deeper understanding of these relationships, providing a foundation for future research aimed at tailoring thin-film photodetectors for specific applications, thereby advancing the field of photodetection technology.

In this study, a significant limitation is our inability to fabricate CdS thin films with thicknesses below 10 nm. This constraint could impact the extrapolation of our findings to ultrathin film applications, where different physical phenomena might become more pronounced. Additionally, the scope of our investigation was confined to specific conditions of thin film synthesis and deposition. While these conditions were carefully selected to optimize the performance of the CdS-based photodetectors, they might not cover the full range of scenarios encountered in diverse practical applications. Recognizing these limitations is crucial for accurately interpreting our results and identifying areas for future research to expand the applicability of our findings.

## 4. Conclusions

In this research, we have systematically investigated the enhancement of photodetection properties in CdS-based devices, achieved through the precise fabrication of CdS thin films utilizing the pulsed laser deposition (PLD) technique. A distinctive aspect of our study is the comprehensive characterization of these films’ morphological, structural, and optical properties, affirming their aptness for high-efficiency photodetection applications. Integrating plasmonic gold nanoparticles (AuNPs) on the surface of CdS films constitutes a significant innovation in our work, leading to marked improvements in device performance, especially in thinner CdS layers. This study pioneers in demonstrating how reducing the thickness of CdS layers can synergistically interact with the surface-deposited AuNPs to substantially enhance key performance parameters, including detectivity, responsivity, external quantum efficiency (EQE), and photocurrent gain. Notably, the configuration of AuNPs with the thinnest CdS layer (S4) exhibited the most pronounced enhancements, underscoring the critical role of nanoscale interactions in photodetector efficiency. A novel contribution of our research is elucidating the surface plasmon resonance (SPR) effect induced by AuNPs, which is pivotal in augmenting the photodetection capabilities of the CdS films.

Further, we provide new insights into how the physical thickness of the CdS layers and the incorporation of AuNPs substantially influence the photodetector’s response and recovery times. Our findings reveal that thinner CdS layers and the presence of AuNPs contribute to faster response and recovery times, parameters crucial for applications requiring rapid detection and response. This aspect of our study demonstrates the technical feasibility and adds to the growing knowledge of optimizing photodetector device response times. The application of the PLD technique for the large-scale fabrication of CdS thin films with surface-deposited AuNPs, as presented in our work, opens new avenues in developing high-performance UV photodetection devices. The methodology proposed herein offers promising prospects for designing and fabricating efficient and responsive photodetectors. Our research significantly contributes to the field of CdS-based photodetectors by emphasizing the influence of film thickness and the incorporation of plasmonic AuNPs on device performance. The presented results contribute to advancing photodetection technology and lay a foundation for future studies. The insights gained from our work provide a valuable framework for further optimization of device parameters and for exploring novel materials and device architectures in photodetection. In conclusion, this study advances the current understanding of CdS-based photodetectors and presents a promising pathway for developing advanced photodetection devices. Future research building upon these findings has the potential to propel the field of photodetection toward new technological frontiers, thereby making a significant impact in both scientific and practical domains.

## Figures and Tables

**Figure 1 nanomaterials-14-00416-f001:**
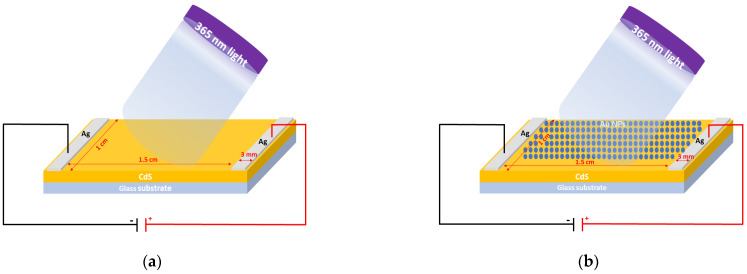
Schematic diagram of the fabricated devices: (**a**) Ag/CdS and (**b**) Ag/AuNPs/CdS.

**Figure 2 nanomaterials-14-00416-f002:**
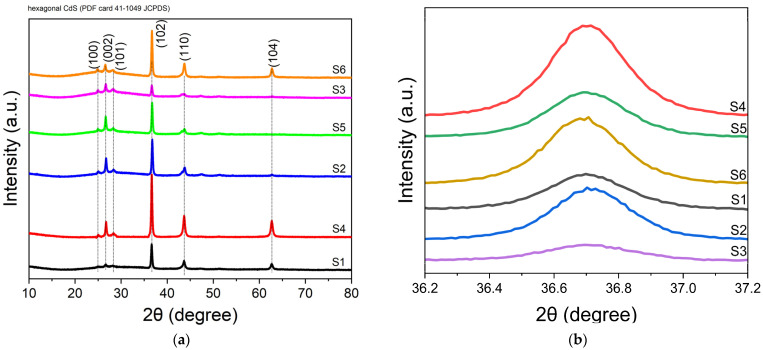
X-ray diffraction (XRD) patterns of samples S1 to S6. (**a**) Overall XRD patterns of samples S1 to S6, showcasing the crystalline structure of each sample. (**b**) Zoomed-in view of the XRD pattern focusing on the (102) direction, highlighting the differences in peak intensity and crystallinity among the samples.

**Figure 3 nanomaterials-14-00416-f003:**
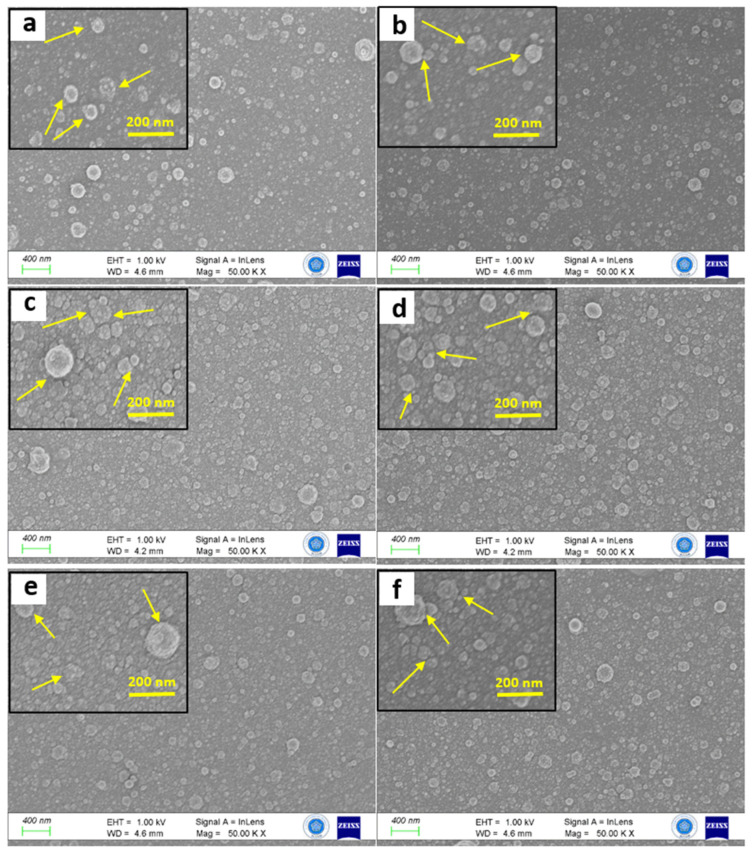
FESEM surface morphologies of samples S1–S6 ((**a**–**f**), respectively). Insets show high-resolution FESEM images at a magnification of 150.00 KX. Arrows indicate the presence of small grains with spherical particles.

**Figure 4 nanomaterials-14-00416-f004:**
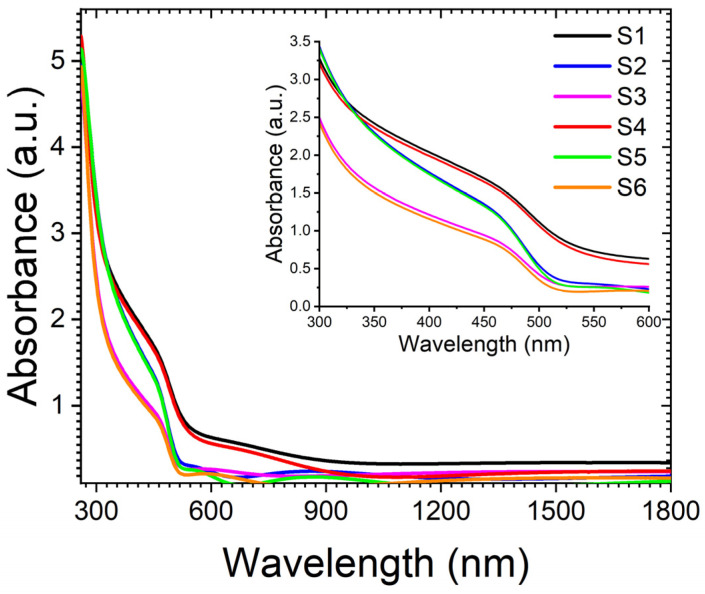
UV–Vis absorbance spectra of samples S1 to S6.

**Figure 5 nanomaterials-14-00416-f005:**
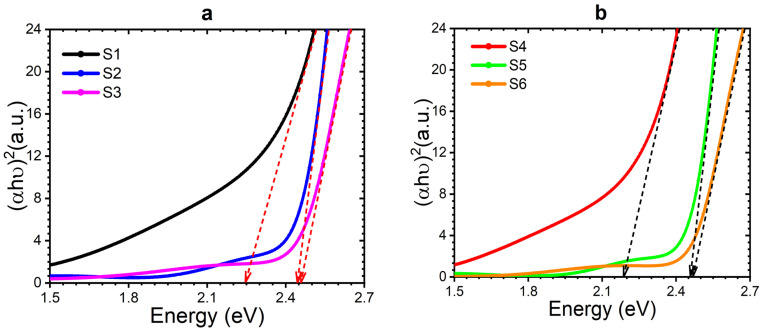
Tauc plots showing band gaps for samples (**a**) S1, S2, and S3, and (**b**) S4, S5, and S6.

**Figure 6 nanomaterials-14-00416-f006:**
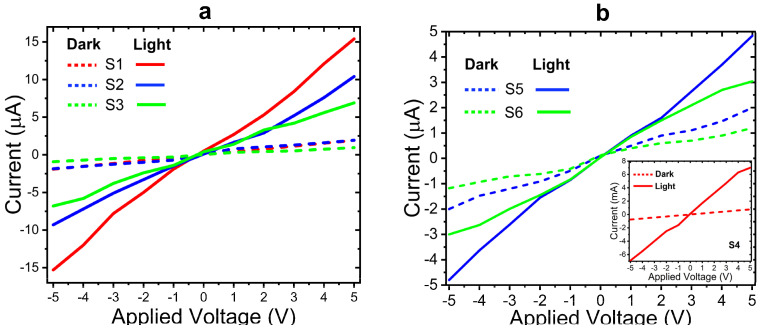
Current–voltage (I–V) behavior of samples (**a**) S1, S2, and S3, and (**b**) S4, S5, and S6.

**Figure 7 nanomaterials-14-00416-f007:**
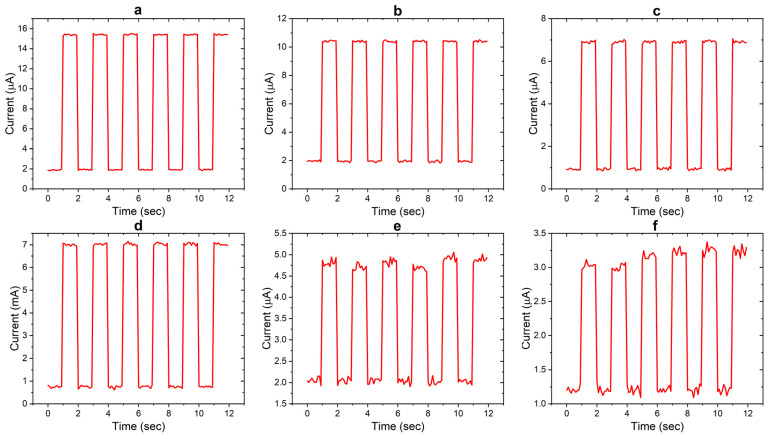
Photo-switching characteristics of samples S1–S6 (labeled (**a**–**f**) respectively).

**Figure 8 nanomaterials-14-00416-f008:**
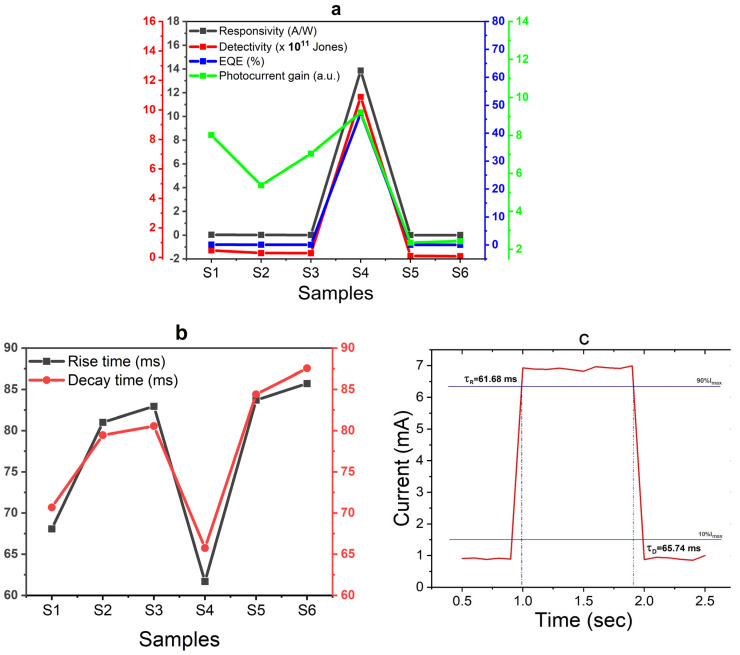
Photodetection parameters of samples S1–S6: (**a**) responsivity, detectivity, EQE, and photocurrent gain, and (**b**) rise and decay times. (**c**) Enlarged view of the temporal photocurrent response for sample S4, illustrating rise and decay times.

**Table 1 nanomaterials-14-00416-t001:** X-Ray diffraction characteristics of samples S1–S6.

Sample	2 Theta (Degree)	FWHM	D (nm)	δ (Lines/m^2^)	*hkl*
S1	24.91	-	-	-	100
	26.65	0.434	19.65 ± 1.78	3 × 10^15^	002
	28.29	-	-	-	101
	36.70	0.377	23.15 ± 0.64	2 × 10^15^	102
	43.77	0.512	17.47 ± 2.69	3 × 10^15^	110
	62.73	0.580	16.75 ± 1.49	4 × 10^15^	104
Mean			19.25 ± 1.29	3 × 10^15^	
S2	24.91	0.539	15.77 ± 1.45	4 × 10^15^	100
	26.65	0.355	24.03 ± 0.12	2 × 10^15^	002
	28.29	0525	16.30 ± 1.84	4 × 10^15^	101
	36.70	0355	24.58 ± 1.70	2 × 10^15^	102
	43.77	0.683	13.09 ± 0.68	6 × 10^15^	110
	62.73	0.603	16.12 ± 0.74	4 × 10^15^	104
Mean					18.31 ± 1.69
S3	24.91	0.617	13.77 ± 1.67	5 × 10^15^	100
	26.65	0.372	22.93 ± 0.30	2 × 10^15^	002
	28.29	0.552	15.30 ± 1.93	4 × 10^15^	101
	36.70	0.428	20.43 ± 2.10	2 × 10^15^	102
	43.77	0.972	9.20 ± 2.17	1 × 10^16^	110
	62.73	-	-	-	104
Mean			16.33 ± 2.02	4 × 10^15^	
S4	24.91	-	-	-	100
	26.65	0.695	12.27 ± 2.20	7 × 10^15^	002
	28.29	-	-	-	101
	36.70	0.298	28.69 ± 1.08	1 × 10^15^	102
	43.77	0.486	18.40 ± 2.17	3 × 10^15^	110
	62.73	0.538	18.07 ± 0.38	3 × 10^15^	104
Mean			19.36 ± 1.85	3 × 10^15^	
S5	24.91	0.501	16.96 ± 0.23	3 × 1015	100
	26.65	0.354	24.09 ± 0.37	2 × 1015	002
	28.29	0.578	14.81 ± 1.28	5 × 1015	101
	36.70	0.382	22.85 ± 0.65	2 × 1015	102
	43.77	0.578	11.15 ± 1.34	8 × 1015	110
	62.73	0.788	-	-	104
Mean			17.97 ± 0.16	3 × 10^15^	
S6	24.91	0.486	17.39 ± 2.24	3 × 10^15^	100
	26.65	0.394	21.65 ± 1.61	2 × 10^15^	002
	28.29	0.565	15.15 ± 0.99	4 × 10^15^	101
	36.70	0.350	24.94 ± 0.24	2 × 10^15^	102
	43.77	0.565	15.83 ± 1.07	4 × 10^15^	110
	62.73	0.551	17.64 ± 2.04	3 × 10^15^	104
Mean			18.77 ± 1.22	3 × 10^15^	

**Table 2 nanomaterials-14-00416-t002:** Comparison of device performance: current device vs. previous studies on CdS thin film-based photodetectors.

Device	λ (nm)	R (A/W)	$D^{*}$(Jones)	EQE (%)	$\tau_{R}$ (ms)	$\tau_{D}$ (ms)	Refs.
Current study (S4)	365	13.86	1.1 × 10^12^	47.19	61.68	65.74	This work
CdS thin film	532	18.8	20.9 × 10^10^	61	200	500	[70]
5 wt.% CdS: Pr thin film	532	2.71	6.9 × 10^10^	628.86	90	170	[45]
1 wt.% CdS: Sm thin film	532	1.01	2.21 × 10^12^	256	138	120	[46]
5 wt.% CdS: Eu thin film	532	0.614	1.38 × 10^12^	143	85	106	[9]
CdS core-Au/MXene	254	8.6 × 10^−2^	1.34 × 10^11^	-	>10^3^	>10^3^	[71]
PPy/CdS QDs	850	3.8 × 10^−3^	2.1 × 10^16^	560	-	-	[72]
CdS: Ga/Au/SiO_2_/Si	510	8	-	-	0.095	0.29	[73]
CdS nanobelts	488	2 × 10^2^	-	520	0.137	0.379	[74]
CdS nanowires	470	0.43	2.58 × 10^11^	-	300	400	[75]
CdS: Ag/Si thin film	551	0.43	2.58 × 10^10^	91.42	-	-	[76]
p-Si nanowires/n-CdS	900	0.82	1.21 × 10^12^	-	203	429	[77]
CdS thin film	405	4.21	-	1.29 × 10^12^	267	277	[17]

## Data Availability

The data presented in this study are available on request from the corresponding authors.

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
