# Peer review of "Utilizing Gold Nanoparticle Decoration for Enhanced UV Photodetection in CdS Thin Films Fabricated by Pulsed Laser Deposition: Exploiting Plasmon-Induced Effects"

_nanomaterials, 2024, doi:10.3390/nano14050416_

Round 1
Reviewer 1 Report
Comments and Suggestions for Authors
This article deals with the study of photodetection properties of CdS thin films and the influence of surface-deposited gold nanoparticles (AuNPs) on their performance. CdS thin films of different thickness were deposited by pulsed laser deposition (PLD) technique on glass substrates after a CdS buffer layer. Morphological structural, and optical properties of those thin films were then investigated. Photodetector devices based on CdS and AuNPs/CdS films were realized, and their performance parameters were evaluated under 365 nm light illumination. The reduction of the CdS layer thickness resulted in enhanced performance characteristics, related to detectivity, responsivity, external quantum efficiency (EQE), and photocurrent gain. Furthermore, AuNP deposition on the surface of CdS films influences the devices with thinner CdS layers. Among the configurations, one sample has exhibited the highest values in all evaluated parameters, including a detectivity, responsivity and photocurrent gain. Finally, the behavior of the photovoltaic cell as a function of the operating temperature, was investigated.
The manuscript needs to be revised. The following main points need to be carefully addressed before it can be considered for publication:
1. First, the novelty of the paper within the large literature in the field should be clearly addressed in the manuscript.
2. In the introduction, among the other deposition methods for preparation of high quality CdS thin film, the chemical vapour deposition and metal organic chemical vapour deposition techniques with the following relevant papers [DOI: 10.1016/0022-0248(92)90076-U; DOI: 10.1016/S0960-8974(97)00031-4] deserve to be mentioned.
3. It is necessary to highlight the added value of this article to the research in the field, both at the end of the introduction section and in the conclusions.
4. In the paragraph “2. Experimental Section”, the brand and model of the PLD system, the laser and the UV-VIS spectrophotometer used are missing. Authors should include this information in the paragraph.
5. At the beginning of paragraph “3. Results and discussion”, it is reported that: “According to the XRD pattern shown in the figure …”, it should be pointed that the figure is Fig. 2.
6. In the same paragraph, about the FESEM observations, the parameters used in the measurements (such as the acceleration voltage, the working distance used and so on) should be included.
7. For a clearer presentation, the morphological analysis (SEM observations) should be moved at the beginning of the discussion of the results, before or after the XRD analysis, however before the Raman results.
8. In the Table 1, all the values need to be reported with their errors.
9. Table 1: the crystalline size values present too many significative digits for having been calculated from Eq. (1). In the calculation of the crystallite size, has the Debye-Scherrer broadening been deconvoluted from the instrumental broadening? If not, it should be clearly stated in the text that the reported crystallite size values are a lower-bound estimates. For this reason there is no meaning in having decimal digits for values of estimated crystallite size. All those values should be approximated according to proper experimental errors.
10. Page 5: Authors state that: “The dislocation density and micro-strain of thin films [50, 51] are determined by Eqs. (2) and (3)”. The authors should pay attention that Eq. (2), by definition, gives the CdS crystallite density, not the dislocation density! Dislocation density can only be measured from TEM observations, when present. As column 2 in Table 1 reports values of the average crystalline density: being these values calculated also in this case from Eq. (2), only orders of magnitude must be reported, i.e., without meaningless decimal places (see point 9 above).
11. Eq. (3) is very strange as it seems to suggest that the strain is inversely proportional to the crystallite size (Eq. (1)). The authors pretend to obtain two different parameters (the crystallite size and the strain) from the same experimental quantity: either is one case or the other. Moreover, a strain in the order of 10-3 is very difficult to measure from a single-crystal XRD pattern. The authors should reconsider the entire interpretation on this matter.
12. In the paragraph 3.3 is stated that: “The formation of small agglomerations can be attributed to the fact that the substrate temperature was maintained at room temperature during the deposition. These observations align with similar effects that have been previously reported [30]”. The SEM images in fig. 4 do not show any appreciable difference between them. Which size have the agglomerations? The authors should clearly explain and carefully clarify this point. Furthermore, which “similar effects” have been previously reported in [30]? Why the comparison of the present results is made with a previous paper of the authors Ref. [30] which focuses on a completely different system, i.e. such as the Cu2SnS3 ternary chalcogenide thin film?
13. In the paragraph 3.5 the PL analysis is reported. The related Figure 7 reports the data as a function of the Energy, while all the text is commented in wavelength (peak position and so on). The authors should at least uniform the text by adding the energy value corresponding to each wavelength mentioned.
14. The conclusions should underline the added value of the article compared to the extensive research carried out in the field and highlight in more depth the potential implications of these studies.
15. The English need to be improved: words such as “captured, depicted, subjected” and so on should be replaced with more appropriate ones. It is recommended that the manuscript be reviewed by a native English speaker.
Comments on the Quality of English LanguageThe English needs to be improved. It is recommended that the manuscript be reviewed by a native English speaker.
Author Response
Thank you for your efforts in enhancing the manuscript and your recommendations.

Reviewer 2 Report
Comments and Suggestions for Authors
The authors of the manuscript titled: "Utilizing Gold Nanoparticle Decoration for Enhanced UV Photodetection in CdS Thin Films Fabricated by Pulsed Laser Deposition: Exploiting Plasmon-Induced Effects" present their work on CdS photoresistors supposedly enhanced by Au nanoparticles. The results seem to be put together quite well on the first look, but I have serious concerns about their trustworthiness.
First, there is no photograph of the tested samples. In photoresistors, the layout of electrodes plays a crucial role in the function of the device. There is no information about the electrode layout (interdigital electrodes are standard for photoresistors).
The presented XRD spectra do not seem to be applicable for the Debye-Scherrer equation (there seems to be no appropriate peak widening (see (102) plane diffraction) that would suggest nanostructure forming). There is no mention about which crystalline plane diffraction was used for the caluculation (if (102) - the strongest, I can not see any widening of the diffraciton peak from the presented spectra).
In Raman spectra the signal decreases with increasing thickness of the CdS layer. There is no better expanation for this than that the presented peaks belong to the underlying substrate, since the analytical depth of Raman spectroscopy is significantly higher than the thickness of the CdS films. The same applies for the photoluminiscence results.
The SEM images show no difference with Au nanoparticles added, contrary to what the authors stated.
In UV-Vis spectra the absorption edge is blueshifted with increasing CdS thickness, contrary to what is stated in the maniscript. More significantly, there is no difference in the spectra of pristine CdS layers and the layers decorated with Au nanoparticles. The surface plasmon resonance in metallic nanoparticles is accompanied by significant light absorption in the visible region (usually 550-600 nm for Au nanoparticles). Since there is no hint of this in these spectra, the presence of plasmonic material could therefore be questioned.
The illumination response of the materials in Fig. 9 raises the most concerns. The data seem to be manipulated or maybe outright fabricated. The recovery time of CdS photoresistors is generally much longer than their response time. Even though the graph is low resolution, the noise pattern seems to be repeating in some cases, however, in most cases no noise in the signal can even be seen. These results are therefore highly untrustworthy.
Comments on the Quality of English LanguageModerate changes to english grammar would be in place. Some parts of the text are difficult to understand, many parts of the discussion of the results seem to be just incoherent blabber.
Author Response

(The authors gave the same response as above.)

Reviewer 3 Report
Comments and Suggestions for Authors
[Abstract]
1. "Furthermore, AuNP deposition on the surface of CdS films exhibited a substantial influence, especially on devices with thinner CdS layers" Can author provide quantitative data to make this statement more informative?
[Background]
1. In the first paragraph, please explain concisely (1-2 paragraphs) about the mechanism of UV sensors. Further, it would be more informative to the readers if authors could add example of the UV sensors utilities.
2. Line 79. "Previous investigations have not addressed the influence of film thickness...." which previous investigations authors are refering to?
3. Line 62-63 "Furthermore, thin film formation can take place in various reactive conditions." What are 'reactive conditions' authors referring to here? In the findings, there is no relevant data to have any relevance to this statement. Can author clarify?
[Methods]
1. I am not sure if Nanomaterials used section title "Experimental Section". They usually have it as "Materials and Methods"
2. "2.1" should be "Materials". In the section authors describe all used materials and their quality. How they were purchased should be explained there as well.
3. Please explain how 'thickness' of the film is controlled.
4. Please explain the replication performed in the study. Including the synthesis and the measurement. Please present the data in Mean(SD).
[Results]
1. Can author estimate the crystaliinity based on the XRD data? Please refer to the following study: Rahmi et al. Arabian Journal for Science and Engineering 48, pages159–167 (2023)
2. "...the particles were well distributed across the entire surface." can other analysis confirm this characteristic?
3. Authors should use arrow to indicate the morphologic findings in FESEM images.
[Discussion]
1. According to the publication guideline, it is of important to state the limitation of the study. Please elaborate the limitations (in the last paragraph before conclusion) that hinders the interpretation or generalization of the findings.
2. Please explain the underlying theories on the effect of film thickness to photodetection.
[Conclusion]
Line 382-385. "The proposed approach offers promising prospects for the design and fabrication of efficient and responsive photodetectors, with potential applications in environmental monitoring, industrial processes, and biomedical diagnostics" Do authors think their findings are sufficient to draw this statement? The study has yet evaluated the specific utility of the CdS.
Comments on the Quality of English Language[Minor]
1. Please revise "5cm" to "5 cm" and "5N" to "5 N"
2. Do not use the short form of did not (didn't). Please revise.
Author Response

(The authors gave the same response as above.)

Reviewer 4 Report
Comments and Suggestions for Authors
Authors investigated the photodetection properties of CdS thin films and the influence of surface-deposited gold nanoparticles (AuNPs) on their performance. The CdS thin films were produced by pulsed laser deposition (PLD) technique with 100, 150, and 200 nm thicknesses. A special merit of the work is the use of various characterization methods to study the structural (XRD, Raman), morphological (FESEM), and optical (Spectroscopy, PL, photoelectric) properties of thin films. They demonstrated that the reduction in CdS layer thickness and AuNP deposition resulted in enhanced detectivity, responsivity, external quantum efficiency (EQE), and photocurrent gain.
I suggest to accept the publication. Just suggestions for future work: It would be interesting to study the effect of some annealing, and if possible the effect of thinner layer down to the grain size (app. 30 nm)
Author Response

(The authors gave the same response as above.)

Round 2
Reviewer 1 Report
Comments and Suggestions for Authors
The authors have properly addressed all the issues, therefore this revised versione of the manuscript is now suitable for publication.
Author Response
Thank you for your valuable comments and suggestions concerning our manuscript. Your comments are practical and very helpful
Reviewer 2 Report
Comments and Suggestions for Authors
There is still no photograph of the samples, the quality of the electrodes therefore can not be assessed.
The photoresponse of the samples is dubious. Photoresistor devices, such as the one presented by the authors, can not be compared (in both detectivity and response times) to devices with heterojunctions as the authors try to suggest in their response ref. 2,3. The fast response is not thanks to the CdS material, but due to the presence of the heterojhunction which means the minority charge carriers quickly recombine.
Sample 4 has, somehow, several orders of magnitude lower resistivity when illuminated, but also when in dark (maybe the electrodes are shorted after Au deposition?). The noise in the graphs for S4 is visibly higher than for samples 1-3. How is it then possible the authors state much higher detectivity for this sample?
The question of the Raman and photoluminiscence spectra is not resolved either. The crystalline quality of material is best assessed not by SEM (which really deosnt show any difference), but by XRD , which the authors even performed on their samples. And the XRD did not show such significant differences in crystalline quality that would suggest the difference in Raman and PL intensity.
Author Response

(The authors gave the same response as above.)

Reviewer 3 Report
Comments and Suggestions for Authors
Thank you for addressing my previous comments. I still found some issues in this manuscript before I can recommend for its acceptance.
1. For line 97-99, authors have to cite all references they have read that they think as having "lack of comprehensive studies".
2. Authors calculated the particle size distribution, how can it explain the particle distribution on the surface, as the latter 'distribution' refers to location. A bit confused here, please clarify.
3. In acknowledgement, please declare whether the authors use AI-based services, either to write, translate, or improve the grammar. Please state how the AI was used, and to what extent.
Comments on the Quality of English Language
Please do recheck for possible typing or grammatical errors.
Author Response

(The authors gave the same response as above.)
